# 3D electron-beam writing at sub-15 nm resolution using spider silk as a resist

Nan Qin[1], Zhi-Gang Qian[2], Chengzhe Zhou[1], Xiao-Xia Xia[2✉] & Tiger H. Tao[1,3,4,5,6,7,8,9✉]

Electron beam lithography (EBL) is renowned to provide fabrication resolution in the deep nanometer scale. One major limitation of current EBL techniques is their incapability of arbitrary 3d nanofabrication. Resolution, structure integrity and functionalization are among the most important factors. Here we report all-aqueous-based, high-fidelity manufacturing of functional, arbitrary 3d nanostructures at a resolution of sub-15 nm using our developed voltage-regulated 3d EBL. Creating arbitrary 3d structures of high resolution and high strength at nanoscale is enabled by genetically engineering recombinant spider silk proteins as the resist. The ability to quantitatively define structural transitions with energetic electrons at different depths within the 3d protein matrix enables polymorphic spider silk proteins to be shaped approaching the molecular level. Furthermore, genetic or mesoscopic modification of spider silk proteins provides the opportunity to embed and stabilize physiochemical and/or biological functions within as-fabricated 3d nanostructures. Our approach empowers the rapid and flexible fabrication of heterogeneously functionalized and hierarchically structured 3d nanocomponents and nanodevices, offering opportunities in biomimetics, therapeutic devices and nanoscale robotics.

[1] State Key Laboratory of Transducer Technology, Shanghai Institute of Microsystem and Information Technology, Chinese Academy of Sciences, Shanghai, China. [2] State Key Laboratory of Microbial Metabolism, School of Life Sciences and Biotechnology, Shanghai Jiao Tong University, Shanghai, China. [3] Center of Materials Science and Optoelectronics Engineering, University of Chinese Academy of Sciences, Beijing, China. [4] School of Graduate Study, University of Chinese Academy of Sciences, Beijing, China. [5] 2020 X-Lab, Shanghai Institute of Microsystem and Information Technology, Chinese Academy of Sciences, Shanghai, China. [6] School of Physical Science and Technology, ShanghaiTech University, Shanghai, China. [7] Institute of Brain-Intelligence Technology, Zhangjiang Laboratory, Shanghai, China. [8] Shanghai Research Center for Brain Science and Brain-Inspired Intelligence, Shanghai, China. [9] Center for Excellence in Brain Science and Intelligence Technology, Chinese Academy of Sciences, Shanghai, China. ✉email: xiaoxiaxia@sjtu.edu.cn; tiger@mail.sim.ac.cn

Human thrive to harness materials—including both natural and synthetic ones—with modern technologies to find new technological opportunities. Many of these new opportunities and discoveries are inherently grounded in manufacturing innovations. Three-dimensional (3d) manufacturing is particularly intriguing and has recently been investigated intensively in the past two decades. With synergetic progresses in materials development, many applications have greatly benefited from the high-resolution manufacturing of 3d structures and devices at micro-/nanoscale such as microfluidics, refractive/diffractive optics, photonic and mechanical metamaterials[1–13]. However, technical challenges—in terms of applicable manufacturing and materials—become more prominent when the features get smaller, especially reaching the deep nanometer scale (i.e., <100 nm). Resolution, structural stability, and shape accuracy are the key factors. For biomedical applications such as cell scaffolds and therapeutic micro-/nanorobotics, biocompatibility, physiochemical stability, and ease of functionalization of 3d-fabricated structures need to be systematically assessed.

Lithography-based methods (such as mask-less multi-photon lithography, MPL, and mask-projection stereolithography, SL) are capable of creating 3d structures using synthetic polymers (i.e., resins) and natural ones (e.g., keratin, silk fibroin, and sericin, where chemical modifications are usually needed for photosensitivity) as photoresists. However, these photolithographic methods fundamentally suffer from diffraction-limited resolution (~100 nm)[9,12,14–24]. Strategies have been proposed to create pseudo-3d structures at deep sub-diffraction scale, but usually with limited design flexibilities[25–33]. By contrast, electron beam lithography (EBL) and ion beam lithography (IBL) are renowned to provide fabrication resolution in the nanometer range, but the major limitation of these techniques is their incapability of arbitrary 3d nanofabrication[34–36]. For example, Kim et al. reported EBL of 2d periodic nanostructures (i.e., hole and dot arrays with minimum feature size of 30 nm) by using naturally extracted silk fibroin proteins as a resist[34]. We reported that EBL and IBL can be used in combination—a precise alignment between these two nanolithography processes at nanoscale is necessary and technically demanding—to generate simple 2d and 3d nanostructures (i.e., one-layer lines and two-layer blocks with minimum feature sizes of 13 and 50 nm, for 2d and 3d patterning, respectively) in silk proteins but IBL inevitably etches the top layer and tends to induce ion contamination[37]. Despite intensive efforts, current 3d nanomanufacturing technologies suffer from drawbacks including the innate paradox between lithographic resolution and structural complexity as well as restrictions in functionalization. New strategies and innovations in both materials and manufacturing technologies have yet to be explored.

Here we report all-aqueous-based, high-fidelity manufacturing of 3d nanostructures and nanodevices using our developed 3d EBL at a resolution of sub-15 nm with genetically engineered recombinant spider silk proteins as the resist. We show that as-fabricated 3d structures have good mechanical strength and structural complexity at nanoscale. These structures can be functionalized—biologically or chemically—and activated as functional devices with controllability of their shape, motion, and lifetime in space and time domains. As proof-of-concept, we have designed and manufactured a new class of nanorobotics with controlled motion in liquid environments powered by bio-fuels at human physiological levels and light-/pH-/heat-triggered degradation for controlled device lifetime. Our approach can be easily employed and facilely adapted by our peer researchers with commercially available EBL tools with simple modifications for a broad range of applications.

## Results and discussion

**Experimental setup and manufacturing capabilities.** EBL is renowned for fabricating 2d patterns at nanometer resolution thanks to the much shorter DeBroglie wavelength of the electrons compared to photon wavelengths (Supplementary Fig. 1). In order to achieve 3d EBL at the deep nanometer scale, the key is the development of appropriate materials which can be regulated to be cross-linked at controlled different depths by the electron beam and has superior mechanical strengths so as to maintain good structural integrity at nanoscale. In this work, we modified a commercial EBL tool (Hitachi S-4800) to adaptably adjust the accelerating voltages (0.5–10 kV, typically) on demand during the exposure based on different structure geometries (Fig. 1a, Supplementary Fig. 2). Recombinant spider silk proteins are synthesized via genetic engineering as the resist with pure water as the resist developer. Structural transitions between amorphous (water-soluble) and crystalline (water-insoluble) silk with precisely directed energetic electrons are well-defined approaching the molecular level within the protein matrix. The trajectory of electrons, which defines the exposure spot, is regulated by the applied accelerating voltage, which allows 3d nanomanufacturing with extraordinary structure complexity. By this method, we successfully manufactured a wide set of 3d sophisticated nanoarchitectures with different geometries and structural complexities (Fig. 1b–i, Supplementary Fig. 3) with a minimum feature size of 14.8 nm (Fig. 1j) using recombinant spider silk proteins. As-fabricated structures have excellent mechanical and thermal stabilities inherited from spider silk, which can be used directly or serve as polymer masters for replication process of various materials (Fig. 1k).

This approach combines the advantages of MPL for maskless direct 3d writing and EBL for unparallel lithographic resolution. Furthermore, genetic or mesoscopic modification of recombinant spider silk proteins provides the opportunity to embed and stabilize physiochemical and/or biological functions within as-fabricated nanostructures, offering great potential for biological adaptation and integration. Compared to other 3d micro-/nanomanufacturing techniques[4,7–18,20–24,34–42], our approach shows important advances and offers key capabilities within one system (Fig. 1l), including (1) the resolution is significantly improved by order(s) of magnitudes; (2) high strength, high fidelity 3d nanostructures enabled by spider silk resists; (3) facile functionalization homogeneously or heterogeneously, and (4) the entire lithographic process is toxic-free and operates at room temperature, namely, no thermal or laser sintering is required and no hazardous chemicals are used or generated.

**Manufacturing mechanism and material optimization.** To date, EBL represents the smallest and finest practical patterning tool and still remains the method of choice for fabricating nanometer-scale structures. The resist, typically a thin organic polymer film, is altered by the passage of electrons. Every incident electron undergoes multiply inelastic or elastic scattering in the resist layer. Ultimate lithographic resolution is limited by scattering and secondary processes within the resist layer. Historically, high accelerating voltages (≥25 kV) and thin resist layers (≤100 nm) are used in EBL for 2d fine patterning as etching masks or photomasks. In this work, we alternatively investigate the interaction of electrons at low accelerating voltages (≤10 kV) with thick resists (in the order of μm), which is yet less explored and is the key to the success of extending EBL's current capabilities from 2d nanopatterning to 3d nanomanufacturing. Elastic scattering of electrons has relatively large scattering with negligible energy

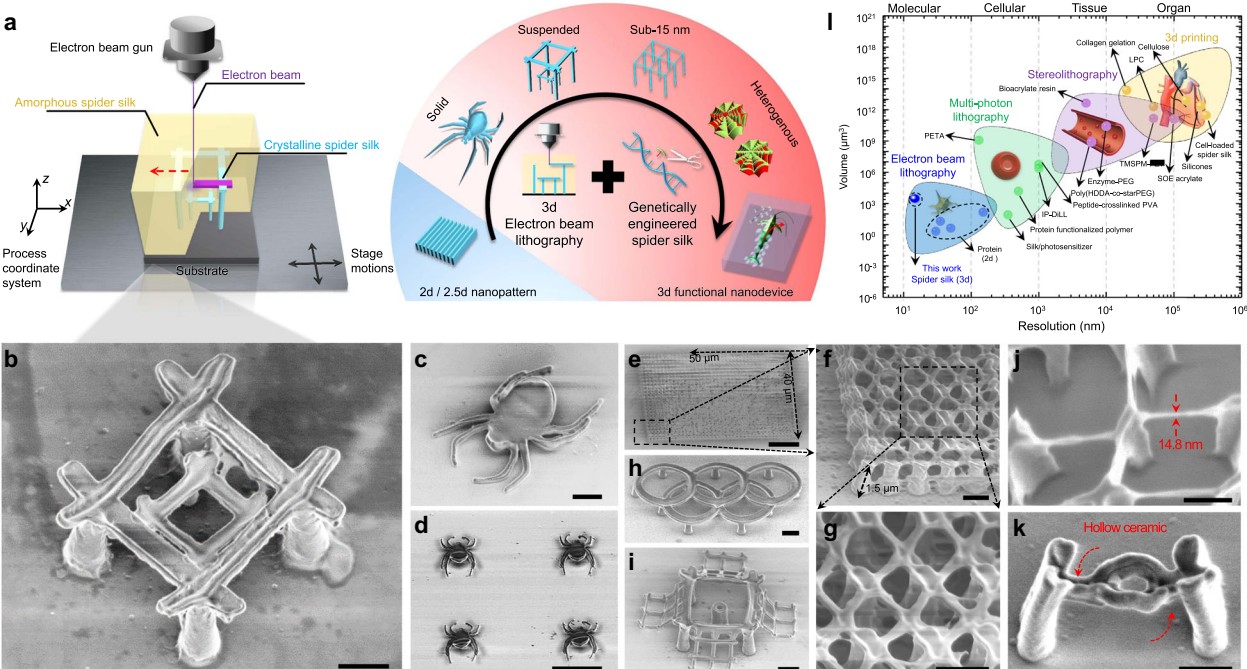

**Fig. 1 Three-dimensional electron beam lithography (3d EBL) using recombinant spider silk proteins as the resist. a** Manufacturing strategy and capabilities of 3d EBL. **b** Nanomatryoshka. **c**, **d** Nanospiders. **e**–**g** Large-scale nanowebs. **h** Nano-Olympic rings. **i** Nanocage. **j** Nanobridge. **k** Single-end anchored/supported hollow ceramic ($Al_2O_3$) nanobeam. Scale bars in the SEM (scanning electron microscope) images: **b**, **c** 1 μm; **d** 5 μm; **e** 10 μm; **f**–**i** 1 μm; **j** 200 nm; **k** 500 nm. **l** The comparison between this work and the existing 3d micro-/nanofabrication methods in terms of resolution and scale. PETA pentaerythritol triacrylate, PVA polyvinyl alcohol, GelMA gelatin methacroyl, SOE-acrylate soybean oil epoxidized acrylate, Enzyme-PEG enzyme-embedded polyethylene glycol, HDDA hexamethylene diacrylate, TMSPM 3-(trimethoxysilyl)propyl methacrylate, PZT lead zirconate titanate, LPC liquid-crystal-polymer.

transfer. In contrast, inelastic scattering of electrons has small scattering angles and transfers (some of) their energy to the resist and results in the resist exposure (Fig. 2a). It is known that that amorphous silkworm and spider silk proteins (water-soluble) can be cross-linked due to the interaction with energetic electrons, and then transforms into the crystalline phase (water-insoluble) for nanopatterning[34,36,37] (Supplementary Fig. 4). Compared to previous studies, the key factor to achieving 3d precise manufacturing in the deep nanometer scale in this work is the unique combination of superior mechanical strengths (key to structure integrity), well-controlled molecular weights (key to high resolution), and, importantly, electron-directed depth-controlled cross-linking of recombinant spider silk proteins (key to 3d direct writing/patterning).

In this context, we first studied the electron–protein interaction in the 3d space by simulating the trajectories of electrons and the energy transfer between electrons and the resist using a customized code modified from the Casino Monte Carlo simulation program[43] (Fig. 2b). It is found that the number of inelastic scattering events and locations, which is highly dependent on the accelerating voltage, defines the effective exposure region within the 3d space of the resist (z direction). Second, we characterized the crystallization of silk proteins after the electron irradiation versus the writing direction and found that the crystallization is highly dependent on the writing direction (x–y direction). This is similar to the natural spinning of spider silks where the crystallization is highly dependent on the spinning direction induced by shear force[44–46] (Fig. 2c–e, Supplementary Fig. 5). By regulating the accelerating voltage thus the exposure depth/region, in addition to moving the electron beam or the sample stage, arbitrary 3d structuring can be readily achieved. Last and importantly, compared to naturally

harvested silk proteins with Gaussian-like distributed molecular weights, laboratory-produced recombinant spider silk has well-defined molecule weight and serves better for high-precision patterning at nanoscale, offering much enhanced lithographic performances—in terms of resolution, contrast, and aspect ratio —than previously reported regenerated silkworm silk proteins[34,36] (Fig. 2f, Supplementary Figs. 6–8).

Larger molecular weight provides higher mechanical strengths but lower resolutions. We synthesized a series of MaSp1 proteins (Major ampullate silk proteins, used for the construction of the frame and radii of orb webs, and as a dragline to escape from predators) with incremental amounts of repeat sequences (i.e., MaSp1 4-mer, 16-mer, 32-mer, and 64-mer) and examined the manufacturing resolution, structural integrity and mechanical strength of 3d EBL using spider silk proteins as the resist. The Young's moduli of the recombinant spider silk proteins increase with the molecular weight, reaching ~11 GPa for the MaSp1 64-mer spider silk (Supplementary Fig. 9), which is consistent with the reported value of natural dragline silk and considerably higher than those of the silk fibroins extracted from cocoons[47]. As shown in Fig. 2g, MaSp1 16-mer spider silk proteins offer a practical balance between resolution and strength for structural integrity and therefore are used throughout this work unless otherwise stated.

**Heterogeneous, hierarchical, biomimetic 3d nanostructures in functionalized recombinant spider silk proteins.** Genetic or mesoscopic modification of recombinant spider silk proteins can impart different functions to the material and as-fabricated 3d nanocomponents and nanodevices. Driven by rational design and ingenuity, silk can be re-engineered from various perspectives and the composition, shape, and function of as-fabricated 3d

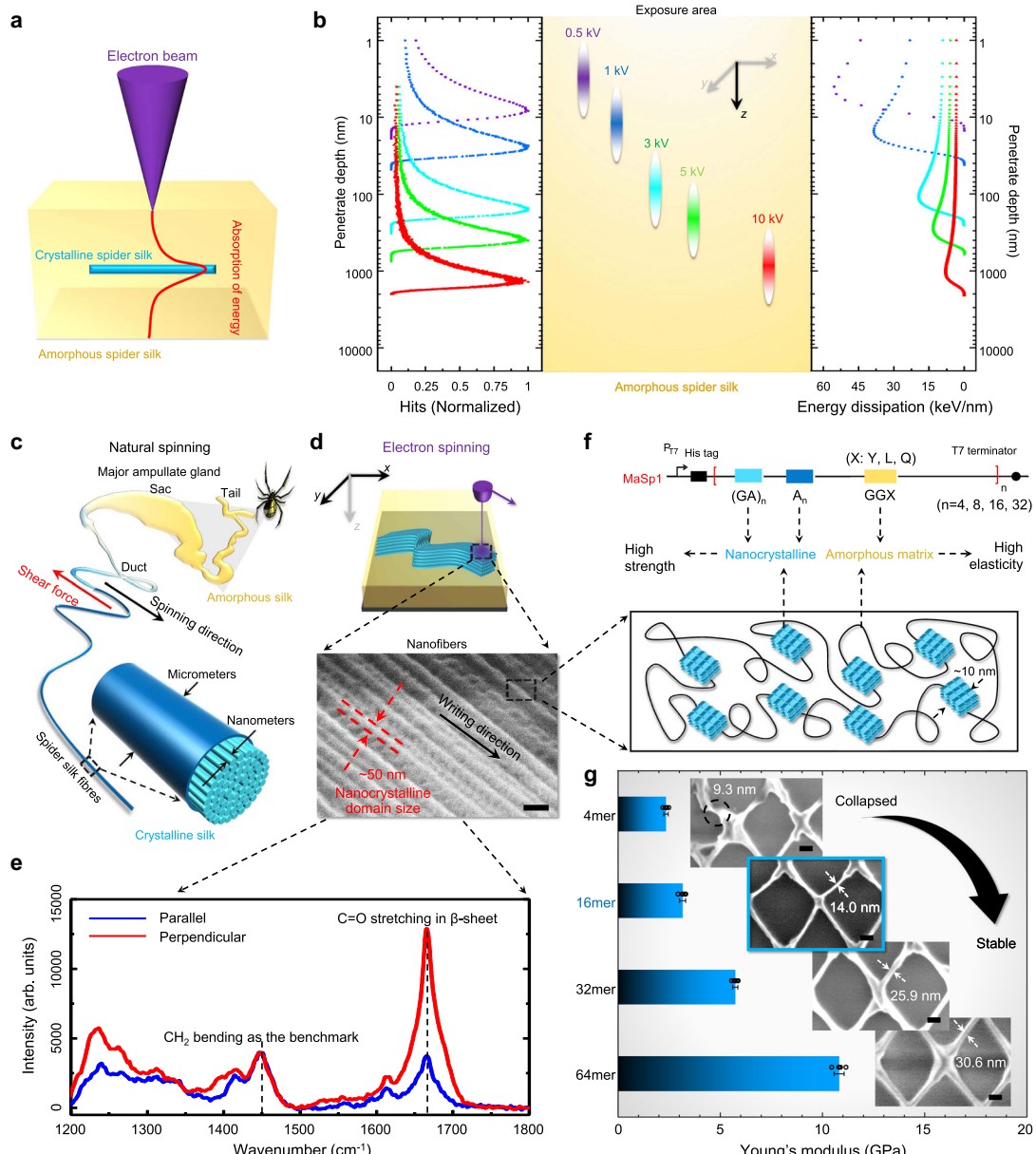

**Fig. 2 Mechanism, material, and manufacturing parameters optimization of 3d EBL in recombinant spider silk proteins. a** Schematic of 3d EBL in the spider silk resist. **b** Monte Carlo simulation of electrons' behaviors (# of electrons $= 10^6$) in spider silk, including the trajectories (left), the energy dissipations (right), and the probable exposure area of electrons at varying accelerating voltages (0.5–10 kV). **c** Illustration of the natural spinning process of spider silk. **d** "Electron spinning" and the SEM image of the formed spider silk protein nanofibers following the writing direction. Scale bar, 200 nm. **e** Polarized micro-Raman spectra of the spun nanofibers. **f** Well-defined molecular structures of genetically engineered spider silk proteins. A: Alanine. G: Glycine. Y: Tyrosine. L: Leucine. Q: Glutamine. **g** Mechanical properties and the minimum feature sizes of 3d EBL using different types of spider silk resists. Scale bars in SEM images (insets), 100 nm. $n = 5$ for each group. The error bars denote standard deviations of the mean.

bionanostructures can be well configured. This offers possibilities to construct heterogeneous, hierarchical nanostructures in functionalized recombinant spider silk proteins fabricated using 3d EBL. For example, different types of silk proteins (e.g., the major ampullate spidroins, MaSp1 and MaSp2) are required for a spider web to gain important biological functions such as adhesive properties (spiral line, sticky and stretchy) and toughness (dragline & radial lines, strong and tough) to catch prey (Fig. 3a, Supplementary Fig. 10). As proof-of-concept, we accurately configured the material distribution and multiscale morphology of a set of biomimetic 3d nanostructures in lateral (x–y) and vertical (z) direction for specific functions under the control of

adjustable cross-linking pathway of energetic electrons in various spider silk layers (Fig. 3b–e). Figure 3c shows the x–y plane controllable configuration of spider nanowebs for PMMA nanospheres (~250 nm in diameter) adhesion. The images in the upper row and lower row were the schematics and the corresponding samples, respectively. We fabricated the radial line (red fluorescence) and spiral line (green fluorescence) in sequence to define the materials and functions of the nanowebs in the x–y plane. When the nanowebs were immersed into the suspension of PMMA nanospheres (~250 nm in diameter), more nanospheres (blue, false color) were adhered to the spiral line (green, false color) than the radial line (red, false color) with a ratio of 34:16.

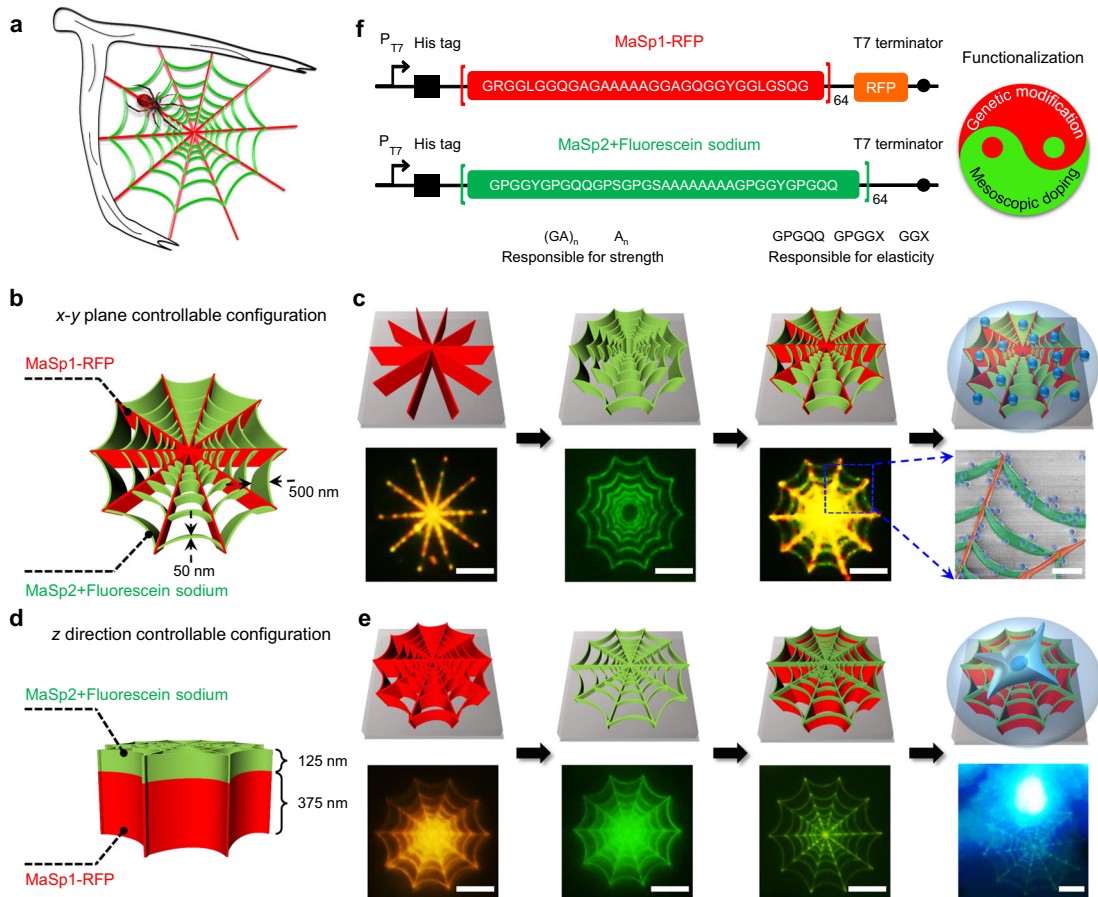

**Fig. 3 Heterogeneous, hierarchical, biomimetic 3d nanostructures using functionalized recombinant spider silk proteins. a** Schematic representation of a natural spider web with heterogeneous and hierarchical structures. **b, c** $x$–$y$ plane controllable configuration of spider nanowebs for PMMA nanospheres (~250 nm in diameter) selective adhesion. Scale bars in (**c**), fluorescence microscope photographs (lower left 3 images), 5 μm; false-color SEM image (lower right), 1 μm. **d, e** $z$ direction controllable configuration of spider nanowebs for single brain tumor cell (~30 μm in diameter) adhesion. Scale bar in (**e**), fluorescence microscope photographs (lower row), 10 μm. **f** Genetically engineered spider silk proteins functionalized with RFP (red fluorescent protein) and fluorescein sodium. R: Arginine. S: Serine. P: Proline.

Figure 3e shows the $z$ direction controllable configuration of spider nanowebs for single brain tumor cell (U251 glioma cell, ~30 μm in diameter) adhesion. As shown in the schematics (upper row), we fabricated the substrate (red fluorescence) and top layer (green fluorescence) of the nanowebs in sequence (lower row) to form a single-cell scaffold with a diameter matching the tumor cell size. Then, the spider silk nanowebs and U251 glioma cells were placed into 6-well plate. After cell-culture as described in the "Methods" section, single brain tumor cell (blue fluorescence) successfully adhered and grew on the nanowebs. By means of precise gene modification on the virgin molecule, the RFP (Red Fluorescent Protein)-labeled MaSp1 64-mer not only inherits the excellent mechanical strength and toughness, but also emits red fluorescence, used as the backbone material of the biomimetic nanostructure. The MaSp2 64-mer with high elasticity was physically mixed with the fluorescent dye sodium fluorescein to accomplish mesoscopic doping, and then used as the adhesion material (Fig. 3f). The RFP and sodium fluorescein were chosen to demonstrate the different means of genetic and mesoscopic functionalization, respectively, and were also used to better illustrate the heterogeneous and hierarchical nanostructures. Notably, the 3d EBL process is friendly to silk proteins as well as these labile molecules embedded within the material for functionalization.

**Biofuel-powered, enzyme-assisted 3d nanorobotics.** Micro-/nanorobotic structures that can be controlled or propelled through chemical or bio-hybrid sources have emerged as exciting options for nano-payload delivery[7,8,11,48–50]. Current micro-/nanorobotic devices usually consist of mechanically distinct components (e.g., propulsion motor, directional thrust, cargo hold of the payload, and delivery system) made of different materials (e.g., various metals and/or polymers for external stimulus-based propulsion). This raises challenges due to limitations of current engineering approaches, particularly in the fields of device fabrication and material science.

The capabilities of creating sophisticated 3d nanostructures with high fidelity and ease of functionalization offer possibilities of developing a class of biomimetic, biocompatible, bioactive, and biodegradable 3d protein-based nanorobots for the target delivery of therapeutic payloads. The nanofishes are functionalized to include three important features within one single device. For device propulsion, two enzymes of glucose oxidase (GOX, glucose → $H_2O_2$) and catalase (CAT, $H_2O_2$→$O_2$) are embedded and stabilized in the 3d-fabricated nanofish for biofuel-powered gas-propulsion in a glucose-containing environment at the human physiological level (Fig. 4a, Supplementary Figs. 11, 12). For directional motion control, different doses of electron irradiation are applied on specific regions of the nanofish to

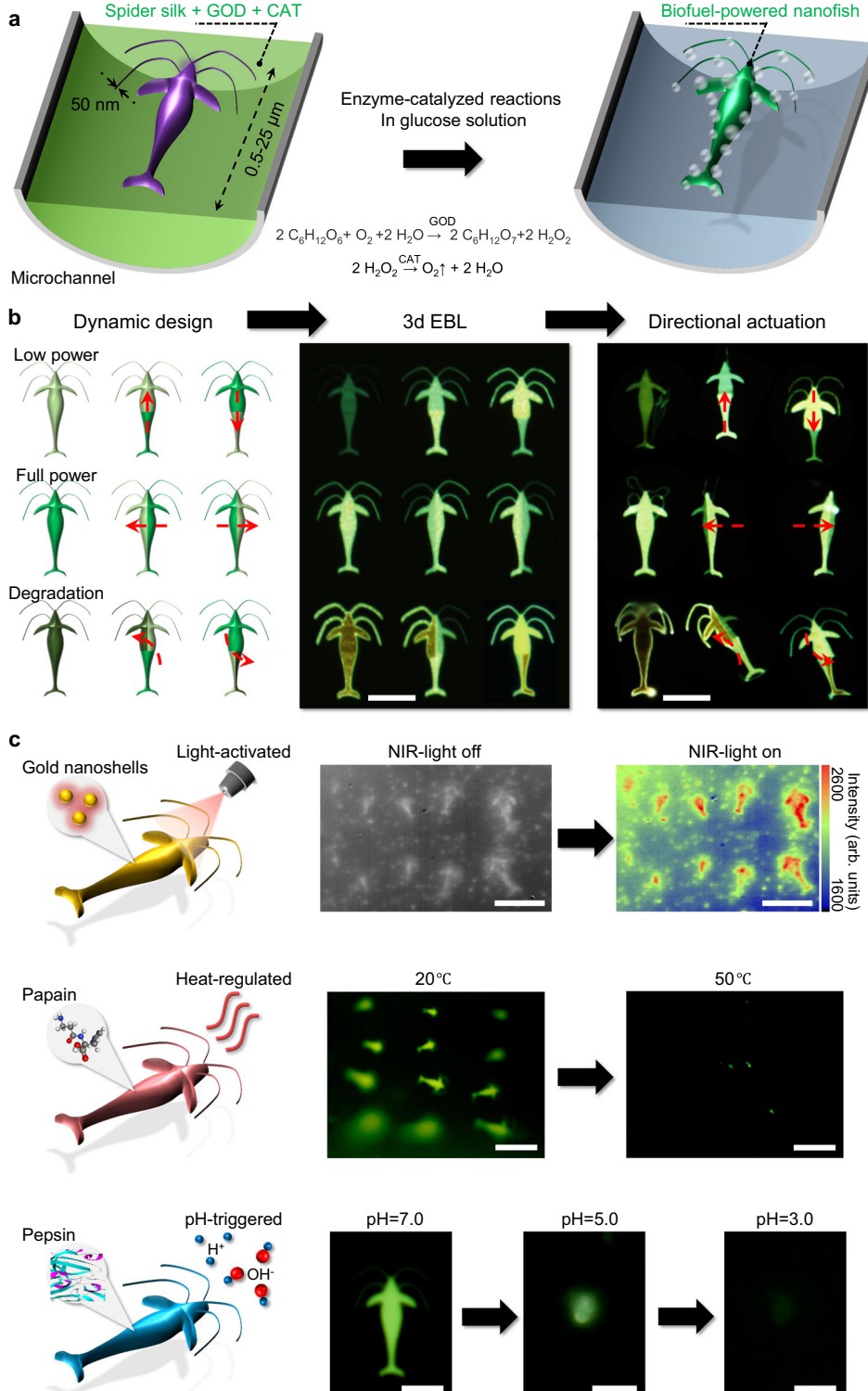

**Fig. 4 Biofuel-powered, enzyme-assisted spider silk nanofishes fabricated using 3d EBL. a** Diagram of the enzyme-assisted glucose-powered nanofishes manufactured in the microchannel. **b** Dynamic design, 3d EBL printed, and the directional actuation of the nanofishes. Scale bars in (**b**), fluorescence microscope photographs (right 2 images), 10 μm. **c** Near-infrared (NIR) light-activated, temperature, and pH-triggered dissolution of the ambient sensing nanofishes for the controllable payload release and device lifetime. Scale bar in (**c**), NIR microscope photographs (first row), 20 μm; fluorescence microscope photographs, 20 μm (second row), and 5 μm (third row), respectively.

adjust the enzyme activities across the device so as to create appropriate force gradients, resulting in different motions (e.g., move up, down, left, right; rotate clockwise or counter-clockwise) (Fig. 4b, Supplementary Fig. 13). The light green, bright green, and dark red fluorescence in nanofishes represented the low power, full power, and loss power states in kinetics. For triggered release, the nanofishes can be facilely loaded with therapeutic molecules together with gold nanoparticles, temperature-sensitive or pH-sensitive enzymes (such as papain or pepsin) for light-activated, temperature-regulated, or pH-regulated drug release, respectively (Fig. 4c). Under the NIR-light illustration, the nanofishes embedded with gold nanoshells (peak absorbance at 980 nm) exhibited stronger infrared absorption than the background. With the increase of the temperature from 20 to 50 °C, papain (pre-loaded in the nanofishes) was activated (optimum temperature 50–60 °C) and led to the degradation of spider silk protein. Pepsin is inactive and active in the neutral and acidic environments, respectively. With the decrease of pH from 7.0 to 5.0 and 3.0, pepsin was gradually activated and broke down spider silk proteins into smaller peptides. It provides opportunities for the controllable drug delivery and release in the digestive systems of humans and many other animals.

**Conclusion**. In summary, we have developed a precise nano-manufacturing process using voltage-regulated 3d EBL to create delicate 3d nanostructures in the deep nanometer scale with high resolution (minimum features at sub-15 nm), structure complexity, and design flexibility. Voltage-regulated electrons define the volumetric energy dissipation and exposure depth within the protein resist. Importantly, recombinant spider silk proteins via genetic engineering—as the resist—offer superior mechanical properties that are essential to building sophisticated nanoarchitectures with high aspect ratios and structural stabilities. Our method using aqueous spider silk resist enables the direct incorporation, immobilization, and stabilization of physiochemical and biological activities in the printed structures at nanoscale. Furthermore, as-fabricated 3d structures can be functionalized and activated for controlled motion and triggered degradation in bio-environments. The fabrication process is compatible with most commercially available EBL tools with simple modifications. Our approach not only allows for the facile, inexpensive, and flexible fabrication of 3d templates for the replication of delicate structures in conventional organic and inorganic materials but also paves the way towards direct production of dynamic functional nanodevices and nanocomponents in bio-friendly and eco-friendly biopolymers, thus offering opportunities in biomimetics, molecular devices, nanoscale robotics and beyond.

## Methods

**Preparation of recombinant spider silk proteins**. The recombinant spider silk proteins were produced by high cell density cultivation of *E. coli* BL21(DE3) transformed with the silk expression plasmids. The cells with soluble expression of silk proteins (MaSp1 4mer, 16mer, 32mer) were lysed by high-pressure homogenization. Following centrifugation, the supernatant was purified by the standard Ni-NTA affinity chromatography. The silk proteins (MaSp1 64mer, MaSp1-RFP, MaSp2 64mer) expressed in an insoluble form were prepared according to previously reported protocols[47]. Cells were suspended and lysed in buffer A (1 mM Tris-HCl, pH 8.0, 20 mM NaH$_2$PO$_4$) supplemented with 8 M urea and 2 M thiourea. The cell suspension was adjusted to pH 4.0 with glacial acetic acid, and fractionally precipitated by adding (NH$_4$)$_2$SO$_4$. The precipitate containing target proteins was solubilised in buffer A containing 8 M urea and dialyzed against water at room temperature for 3 days. Following centrifugation, the supernatant was concentrated using an Amicon® Ultra-15 centrifugal filter unit.

**Preparation of silkworm silk proteins**. Silkworm silk proteins were extracted using the established protocols[36]. *Bombyx mori* cocoons (Sericultural Research Institute, Chinese Academy of Agricultural Sciences, Zhenjiang, China) were boiled for various time durations (e.g., 30, 60, and 120 min) in aqueous 0.02 M Na$_2$CO$_3$ (Sigma-Aldrich, USA) and then rinsed for 3 × 30 min in distilled water to remove the Na$_2$CO$_3$ and sericin. The HTP silk was prepared at 121 °C and 15 psi for 3 h. The degummed cocoons were allowed to dry for more than 12 h and then subsequently dissolved in 9.3 M LiBr (Sigma-Aldrich, USA) solution at 60 °C for 3–4 h. The solution was dialyzed for 2 days in distilled water using Slide-a-Lyzer dialysis cassettes (MWCO 3,500, Pierce, USA). The solution was centrifuged for 2 × 20 min at ~24,000 × *g*. The concentration was determined by measuring a volume of solution and the final dried weight.

**SDS-PAGE analysis**. Protein samples were loaded onto 12% SDS-polyacrylamide gels. The gels were stained with Coomassie brilliant blue R250 (BioRad) and imaged using a Microtek Bio-5000plus densitometer (Shanghai, China). The protein bands were normalized and quantified by Image J 1.52u to estimate purity.

**Sample preparation for electron beam nanostructuring**. The silk protein solution was spin-coated on silicon wafers. The film thickness could be tuned by the concentration of the silk solution and the rotation speed. In general, a 500-nm-thick silk protein film was generated by spin-coating 7% spider silk protein solution at a maximum speed of 3000 r.p.m. for 60 s. Then, the sample was soft-baked on a hotplate at 110 °C for 30 min to remove residual water and enhance the stability. Cross-linked film (i.e., crystalline) was produced by soaking in pure methanol for 15 min. An electron beam nanolithography system (Hitachi-4800, Japan) was used to expose the silk protein thin films and observe the morphology. The exposure dosages varied from 50 to 20,000 μC/cm$^2$ at 0.5–10 kV with a probe current of 1 μA. The exposed samples were then developed by immersion in deionized (DI) water for 30–300 s and dried at ambient condition.

**Polarized micro-Raman spectroscopy**. Micro-Raman spectrum of the nanofibers fabricated using electron beam was obtained using a Renishaw Raman microscope (UK). The 632.8 nm red line of a He-Ne laser was focused down to a diameter of ~1 μm and gave 5 mW of energy at the surface of the sample. The data were collected using a ~1000 cm$^{-1}$ spectral window. Raman spectra were recorded as the intensity changing with the wavenumber. Each spectrum takes the form of a background and a series of peaks. No sign of structural deterioration was observed during the measurement period.

**Infrared nanospectroscopy using AFM-IR**. NanoIR system (Anasys Instruments, CA, USA) was used to provide the high spatial and spectral resolution IR absorption information using a combination of AFM and IR laser source. Due to the absorption of the IR laser, local thermal expansion of the sample was detected and thereby mapped as a function of wavenumber. Before the IR spectra measurement of the point of interest, topography images were scanned. The wavelength of the collected IR spectrum was ranging from 1490 and 1690 cm$^{-1}$, with a spectral resolution of 1 cm$^{-1}$ using multi-region laser power settings to ensure consistent signal-to-noise ratio. Ten repeated scans on the same spot and five measurements on adjacent spots were averaged. The protein sample data was normalized with respect to that of silicon under the same conditions, including room temperature (25 °C) and humidity (about 18% relative humidity). Subsequently, 10-point smoothing algorithm was carried out on each data.

**Decomposition of the amide I band**. The band decomposition was performed with the OPUS software package (version 4.2) supplied by Bruker™. As a starting point for the curve-fitting procedure, four individual absorption bands were proposed at ~1620, 1640, 1658, and 1670 cm$^{-1}$, defining β-sheets, unordered random coils, α-helix and β-turn structures, respectively[36,37]. The curve was successfully fitted using the damped least squares optimization algorithm developed by Levenberg-Marquardt and assuming Gaussian band envelopes.

**Nanoscale mechanical properties test**. A Multimode VIII scanning probe microscope (Bruker, USA) was used to measure the mechanical properties of silk proteins at nanoscale, operating in tapping mode under ambient conditions at a scan rate of 1 Hz. The microscope was protected with an acoustic hood to hinder vibrational noise. The AFM cantilevers (Bruker, USA) were calibrated using the reference samples. The data analysis of the Derjaguin–Mueller–Toporov (DMT) modulus was performed by the software Nanoscope Analysis.

**Monte Carlo modeling**. We carried out computation simulation[51] with Casino v2.51 (University of Sherbrooke, freely available) to investigate the 10$^6$ electrons behaviors in the 1-μm-thick spider silk layer coated on a silicon substrate at different accelerating voltages of 0.5–10 kV. The Monte Carlo simulation is a widely used numerical technique for studying scattering process. It traces individual particles, following paths that have been stochastically determined based on the microscopic phenomena and effects that have been modeled theoretically. CASINO is a well-established Monte Carlo code specializing in simulating electron trajectories and the distribution of the backscattered and the absorbed energies in the sample.

The core of the CASINO program is the Monte Carlo method, which simulates the electron trajectories by randomly sampling collision events according to precalculated values of partial elastic cross-section. It is assumed that the diffusing

atom is represented by Rutherford's shielded potential, and the deviation of the trajectory of the penetrating electron at impact is obtained from the differential cross-section of the elastic scattering. The energy losses experienced by the penetrating electron along its entire trajectory until its termination or escape are grouped into a continuous energy loss function.

The program offers a user-friendly graphical user interface that can easily set specific conditions for the simulation of the exposure process during electron beam lithography. It is helpful and common to perform simulation experiments in CASINO, which provides guidance to the optimum process parameters, thus reducing the trial number of expensive, real experiments. In this work, each stimulation took about 10 h to complete. The size of raw data output from CASINO ranges from 3 to 10 gigabytes, depending on the total electron number and the incident beam energy of the individual simulation.

Although CASINO is an outstanding program in electron beam simulation, it only provides basic statistical tools for analyzing the raw data, mostly limited to one-dimensional histograms in depth/radial distance and two-dimensional contour maps of volume slices. It is not possible to perform customized histogram or statistical variables in CASINO. For our purpose, the distribution of energy loss along the electron trajectories is what's most relevant, however, not natively available in the program. On the other hand, compression of dimensionality inevitably causes loss of spatial details in the three-dimensional Monte Carlo simulation, which are vital to exploring 3d electron beam sculpting at nano-scales. What differentiates the work here from most users of CASINO is that, our post-process of the raw data from CASINO (which contains the information of $10^6$ trajectories) directly analyzes the three-dimensional statistics of the Monte Carlo simulation which was not systematically examined before. As a result, in addition to the (low-dimensional) electron energy distribution or trajectories commonly reported in the literature, our analysis highlights the spatial distribution of energy dissipation, a quantity that directly characterizes electron-specimen interaction during the scattering process of electron beam exposure. The post-processed data demonstrated in this work provides a plausible explanation for the penetration effect observed in our experiments.

We briefly explain the methodology behind depth histogram shown in Fig. 2b. A typical electron trajectory $s^{(i)} = \mathbf{x}_0^{(i)}, \ldots, \mathbf{x}_{N-1}^{(i)}$ from CASINO's raw data is identified with $N$ nodes where $\mathbf{x}_k^{(i)} = (x_k^{(i)}, y_k^{(i)}, z_k^{(i)})$ is the coordinates at which the $k$-th elastic scattering event takes place. An electron trajectory terminates when its remaining energy reaches below a minimum threshold, usually around 0.05 keV where the physical models assumed in CASINO no longer hold. A depth histogram is an approximation to the continuous distribution of various quantities of interest through the process of registering events among corresponding depth bins.

Let $\Delta Z$ be the uniform bin size of a depth-histogram $H$ over a depth interval $[0, z_{max}]$ and $Z_m = m\Delta Z$ be the lower boundary of the $m$-th bin. We introduce a bin function $b(z)$ which maps depth value $z$ to some unique bin index $i$ such that $z$ falls into the $i$-th bin interval $I_i = [Z_i, Z_{i+1}]$. A histogram bin $h_i$ holds various statistics that occur inside interval $I_i$.

A z-max (ZM) histogram concerns the statistics of the maximum depth that each electron trajectory $s^{(i)}$ reaches. Mathematically, it is defined as

$$z_{ZM}^{(i)} = \underset{k}{\arg\max}\, z_0^{(i)}, \ldots, z_k^{(i)}, \ldots, z_N^{(i)} \qquad (1)$$

The corresponding bin $h^{ZM}$ of index $b(z_{ZM}^{(i)})$ receives one hit. Looping over all electron trajectories concludes the ZM histogram.

Dissipation statistics is more involved. Since CASINO lumps all energy dissipation into a continuous loss between adjacent nodal positions of an electron trajectory, care must be taken when dealing with statistics of energy dissipation. Let $\Delta e_k^{(i)}$ denote the difference between electron energies at $\mathbf{x}_{k-1}^{(i)}$ and $\mathbf{x}_k^{(i)}$. We adopt a zeroth-order approximation which uniformly distributes dissipation $\Delta e_k^{(i)}$ to the minimal set of consecutive bins that enclose these two nodes. For total dissipation (TD) histogram $H^{TD}$, each of the intermediate bins receives identical amount of dissipation whereas the first and the last bin receive a fraction, depending their actual depth in the bins. Each of these bins also accumulates one hit in the dissipation count (DC) histogram $H^{DC}$ due to its participation in energy dissipation. It is convenient to define the weight factors

$$w = \begin{cases} \dfrac{|\Delta Z|}{|[z_{k-1}^{(i)}, z_k^{(i)}]|}, & \text{Intermediate bins} \\[2ex] \dfrac{|[z_{k-1}^{(i)}, z_k^{(i)}] \cap I_{b(z_{k-1}^{(i)})}|}{|[z_{k-1}^{(i)}, z_k^{(i)}]|}, & \text{First bin} \\[2ex] \dfrac{|[z_{k-1}^{(i)}, z_k^{(i)}] \cap I_{b(z_k^{(i)})}|}{|[z_{k-1}^{(i)}, z_k^{(i)}]|}, & \text{Last bin} \end{cases} \qquad (2)$$

We then update bin statistics according to the weights: $h_m^{TD} \rightarrow h_m^{TD} + w\Delta e_k^{(i)}$ and $h_m^{DC} \rightarrow h_m^{DC} + 1$.

In summary, the above procedures yield three z-histograms: z-max $H^{ZM}$, total dissipation $H^{TD}$, and dissipation count histogram $H^{DC}$. The average dissipation histogram $H^{AD}$ is derived from $H^{TD}$ and $H^{DC}$ through bin-wise division, i.e., $h_m^{AD} = h_m^D/h_m^{DC}$ for each bin index $m$. However, since the presence of electron trajectories drastically drops as a certain depth threshold is reached, the average

dissipation histogram $H^{AD}$ inevitably encounters the issue of division by zero. Even for bins of non-empty data, low statistics could cause large variance in the estimated value of average dissipation at large depth. We tackle this problem by (1) using a larger number (e.g., $10^6$ used in this work) of incident electrons for Monte Carlo simulation and (2) averaging $H^{AD}$ from multiple (e.g., 10 times) but identical simulations to reduce data variance. In the end, $H^{AD}$ gradually converges to a terminal shape as the number of repetitive simulations increases, which reveals the existence of dissipation-per-electron peaks. We also apply a similar variance-reducing procedure to $H^{TD}$ and $H^{DC}$ histograms.

**Cell culture and imaging.** Thanks for the kind help from Prof. Hongying Sha (State Key Laboratory of Medical Neurobiology, Fudan University, China). U251 glioma cells were cultured in the medium of DMEM + 10% fetal serum and transferred to 6-well cell-culture plate (Corning). There were about $5 \times 10^4$ cells in each well. The 3d nanostructures of spider silk protein were immersed in culture solution. The cell-culture plate was placed in an incubator at a temperature of 37 °C, and regulated with 5% $CO_2$ for another 2 d. For immunofluorescence staining, primary antibodies were diluted as follows: human anti-Nesting 1:400 (Abcam, UK), human anti-GFAP 1:500 (Abcam, UK). Secondary fluorochrome-conjugated antibodies were diluted 1:1000 (mouse anti-human, Abcam, UK). Spider silk protein nanowebs and U251 glioma cells were mounted with slides after the treatment with 0.25 µg µL$^{-1}$ DAPI (Sigma, USA) for fluorescence counter-staining of cell nuclear. Cell imaging was obtained using a fluorescent microscopy (Carl Zeiss A1, ZEN lite 3.4, Germany) with a magnification of ×40. The sample was excited from the top surface by an excitation source.

**Biomimetic "preying" with 3d functional spider nanowebs.** The sequential gene expression in cells generates the biological product (i.e., proteins) with unique properties that human need. The major ampullate spidroins, MaSp1 and MaSp2, are well known of their superior mechanical properties. RFP (Red Fluorescent Protein) is found in the Discosoma sp. of a sea anemone and has been widely used in eukaryotic cells, such as animals, plants, and yeasts, to report gene expression due to the strong capability of penetrating the tissue. Benefiting from the precise genetic engineering at DNA level, the RFP-modified MaSp1 (64-mer) was bio-synthesized and found to exhibit excellent mechanical strength and emit red fluorescence. This chimeric spidroin was hence selected as the backbone material for creating the biomimetic nanowebs. The MaSp2 (64-mer) with high viscosity and elasticity was physically mixed with the fluorescent dye sodium fluorescein for mesoscopic doping, and then used as the cell bioadhesion structure. The materials, shapes, and functions of the 3d nanowebs can be configured on demand in lateral ($x$–$y$) and vertical ($z$) direction. Biomimetic capturing using 3d heterogeneous and hierarchical nanowebs had been successfully achieved to "prey" on PMMA nanospheres (polymethyl methacrylate, ~250 nm in diameter) and single brain tumor cell (U251 glioma cell, ~30 µm in diameter).

**Fabrication of spider silk nanofishes.** Catalase (from bovine liver, lyophilized powder 2000−5000 U mg$^{-1}$) and glucose oxidase (from Aspergillus niger lyophi-lized powder 100 U mg$^{-1}$) was obtained from Sigma-Aldrich. A mixture of catalase and glucose oxidase with a weight ratio of 1:3 was dissolved in 500 µL of spider silk aqueous solution. The exposure dosages varying from 500 to 20,000 µC cm$^{-2}$ at 0.5–10 kV were used to define the dynamic distribution of the nanofishes. The concentration of glucose (Sigma-Aldrich, USA) solution was about 5–10 mM. Gold nanoshells (peak absorbance @ 980 nm, 0.1 mg mL$^{-1}$, Sigma-Aldrich), papain (0.1 mg mL$^{-1}$, Sigma-Aldrich), and pepsin (0.1 mg mL$^{-1}$, Sigma-Aldrich) were added into the spider silk solution to achieve the light-activated, heat-regulated and pH-triggered degradation of nanofishes, respectively. The NIR microscope images were collected using LEICA/VISTEC INM 100 (Germany).

**Statistics and reproducibility.** Each experiment was repeated at least three times independently. The experimental outcomes between independent experiments were in all cases comparable. All data are presented as mean ± standard deviation.

**Reporting summary.** Further information on research design is available in the Nature Research Reporting Summary linked to this article.

## Data availability

All data needed to evaluate the conclusions in the paper are present in the paper and the Supplementary Information. Additional data related to this paper may be requested from the authors. The computational data have been deposited in Zenodo at https://zenodo.org/record/5112635.

## Code availability

The Monte Carlo simulation was conducted using Casino v2.51 (University of Sherbrooke, freely available). Replication code[51] is available from GitHub at https://github.com/TigerHTaoLab/CASINO-parser.

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

## Acknowledgements

This work was partially supported by National Science and Technology Major Project from the Minister of Science and Technology of China (grant nos. 2018AAA0103100, 2020AAA0130100, 2020YFA0907702, and 2019YFA0905200), National Science Fund for Excellent Young Scholars (grant no. 61822406), National Natural Science Foundation of China (grant nos. 61574156, 51703239, 31470216, 21674061, and 22075179), Shanghai Outstanding Academic Leaders Plan (grant no. 18XD1404700), Shanghai Sailing Program (grant nos. 17YF1422800), Key Research Program of Frontier Sciences, CAS (grant no. ZDBS-LY-JSC024) and the Scientific Instrument Developing Project of the Chinese Academy of Sciences (grant no. YJKYYQ20170060). We thank Y. Mao, L. Chen, Z. Shi, and Q. Tang from Huashan Hospital of Fudan University in Shanghai for their assistance with cell experiments.

## Author contributions

T.H.T. and N.Q. conceived the idea. N.Q., Z.Q., and X.X. prepared the samples. T.H.T. and N.Q. performed the experiments, T.H.T., N.Q., and C.Z. carried out the simulations, and analyzed the data. T.H.T., N.Q., and X.X. prepared the manuscript. All authors discussed the results and commented on the manuscript.

## Competing interests

The authors declare no competing interests.

**Additional information**

