## [Peer Review File · Nature Communications]

REVIEWER COMMENTS

Reviewer #1 (Remarks to the Author):

The manuscript by N. Qin et al. reports a new pathway to demonstrate 3-dimensional biomaterial nanostructures using the engineered silk protein and the electron-beam lithography method. I think this work shows a remarkable advance for nanostructuring biomaterials in two aspects: (1) expanding the silk e-beam lithography to 3D and (2) using smaller building protein blocks with reduced molecular weight. The manuscript is well-written and also addresses interesting applications. However, there are addressable points authors need to consider. I will be able to recommend its publication after resolving the below concerns.

(1) I cannot understand the fabrication method clearly. In the schematic, high-energy electrons induce cross-linking of the amorphous silk layer. After the exposure, how do you develop the sample? I could not find the description for the development process. The development process is important in lithography, and therefore should be explained.

(2) In the paper by S.Kim et al., it says that electrons make alpha-helix structures instead of beta-sheets. However, micro-Raman data in this work reveals the change of the beta-sheet content. For clear understanding, another analysis such as FTIR is required to show molecular transition happened by electron beams.

(3) Swelling issue during the development process is a serious concern in nano-lithography. It is necessary to address the effect of swelling on minimum feature sizes.

(4) In Fig. 2a-c, it is true that a higher energy beam can penetrate deeper. However, I cannot understand why e-beam hits and energy dissipation have maxima, not exponential decays, with dependences on the e-beam energy. Did the authors shift the focal points as changing the acceleration voltages? It requires a clear explanation on it.

(5) Detailed descriptions for Fig. 4 are required.

(6) In Fig. 4c, NIR-light off image is not comparable to NIR-light on image. Are two the same image?

Reviewer #2 (Remarks to the Author):

This manuscript reports fabrication of 3D structures via voltage modulated electron beam lithography of silk-based films. The dependence of the penetration depth on the acceleration voltage forms the basis of such patterning approach. Functional nanoscale structures can be obtained by modulating the silk-based resists. The work is certainly interesting and can be considered for publication in Nature Communications. The authors should address the following issues.

1) An important metric for lithography is the minimum pitch that can be fabricated. Please report the minimum pitch that can be fabricated with this approach. It will be highly useful to see an array

of pillars at such pitch values.

2) It appears that there is certain degree of line edge roughness in the fabricated structures. Please report line edge roughness values for both in plane and out of plane structures.

3) For a pillar like structure, what is the highest achievable aspect ratio?

4) Some sections (brain tumor cells, images in part c and e) of Figure 3 are not adequately described/discussed in the text.

5) Similarly, some sections (NIR activation, heat and pH driven transitions) of Figure 4 are not adequately described/discussed in the text.

6) What is the achievable level of velocity for the nanofish?

We sincerely thank the editors and reviewers for the consideration and invaluable comments. We went through two reviewers' comments thoroughly, and we respectfully agree with those insightful comments and helpful advice raised by the reviewers. We have carried out additional experiments and revised the manuscript accordingly. All changes have been highlighted in yellow in the main text and supplemental information as needed. We are confident that we have answered and addressed all the comments received and that the manuscript is suitable for publication in Nature Communications. We will remain available to address any other concerns or changes if necessary.

We look forward to hearing back from you.

All the best,

Tiger H. Tao on behalf of all authors

Responses to the referees' comments: Manuscript ID NCOMMS-21-09897

Responses to the comments of Reviewer #1:

The manuscript by N. Qin et al. reports a new pathway to demonstrate 3-dimensional biomaterial nanostructures using the engineered silk protein and the electron-beam lithography method. I think this work shows a remarkable advance for nanostructuring biomaterials in two aspects: (1) expanding the silk e-beam lithography to 3D and (2) using smaller building protein blocks with reduced molecular weight. The manuscript is well-written and also addresses interesting applications. However, there are addressable points authors need to consider. I will be able to recommend its publication after resolving the below concerns.

Response: Thanks for the positive comments.

(1) I cannot understand the fabrication method clearly. In the schematic, high-energy electrons induce cross-linking of the amorphous silk layer. After the exposure, how do you develop the sample? I could not find the description for the development process. The development process is important in lithography, and therefore should be explained.

Response: Thank you for your suggestion. We have added the description for the development process to the Methods section: Sample preparation for electron beam nanostructuring. **The exposed samples were then developed by immersion in deionized (DI) water for 30-300 s and dried at ambient condition.** (Page 12, line 314)

(2) In the paper by S. Kim et al., it says that electrons make alpha-helix structures instead of beta-sheets. However, micro-Raman data in this work reveals the change of the beta-sheet content. For clear understanding, another analysis such as FTIR is required to show molecular transition happened by electron beams.

Response: Thank you for your suggestion. We have added the nanoscale infrared spectrum of spider silk protein treated with electron exposure characterized by AFM-IR in Supplementary Figure 5. A further deconvolution of the amide I band was conducted and the secondary structure content of each condition was quantified. **The changes in the characteristic peaks of the spider silk proteins indicates a significant increase in the fraction of beta-sheet regions (~1,620 cm⁻¹) and a decrease of random coil (~1,640 cm⁻¹) and alpha-helix regions (~1,658 cm⁻¹).** The detailed information for AFM-IR analysis have been added to the Methods section. (Page 13, line 324)

Supplementary Figure 5. Nanoscale analysis of conformational transition of spider silk proteins under electron irradiation using AFM-IR (Atomic force microscopy - infrared spectroscopy).

Infrared nanospectroscopy using AFM-IR. NanoIR system (Anasys Instruments, CA, USA) was used to provide the high spatial and spectral resolution IR absorption

information using a combination of AFM and IR laser source. Due to the absorption of the IR laser, local thermal expansion of the sample was detected and thereby mapped as a function of wavenumber. Before the IR spectra measurement of the point of interest, topography images were scanned. The wavelength of the collected IR spectrum was ranging from 1,490 cm^{-1} and 1,690 cm^{-1} , with a spectral resolution of 1 cm^{-1} using multi-region laser power settings to ensure consistent signal to noise ratio. Ten repeated scans on the same spot and five measurements on adjacent spots were averaged. The protein sample data was normalized with respect to that of silicon under the same conditions, including room temperature (25 °C) and humidity (about 18% relative humidity). Subsequently, 10-point smoothing algorithm was carried out on each data.

Decomposition of the amide I band. The band decomposition was performed with the OPUS software package (version 4.2) supplied by Bruker™. As a starting point for the curve-fitting procedure, four individual absorption bands were proposed at ~1,620, 1,640, 1,658 and 1,670 cm^{-1} , defining β -sheets, unordered random coils, α -helix and β -turn structures, respectively^{36, 37}. The curve was successfully fitted using the damped least squares optimization algorithm developed by Levenberg-Marquardt and assuming Gaussian band envelopes.

(3) Swelling issue during the development process is a serious concern in nano-lithography. It is necessary to address the effect of swelling on minimum feature sizes.

Response: Thank you for your suggestion. Swelling issue is indeed a serious concern in nano-lithography. We have employed several strategies to reduce the effect of swelling on minimum feature sizes. Firstly, varying exposure dosages were applied to adjust the crosslinking level of spider silk proteins for **improving the hydrophobicity of nanostructures** with different feature sizes. Secondly, **development time in water for 3d nanoarchitecture with different complexity was well-defined** to avoid the swelling caused by the diffusion of water molecules into the 3d matrix of crosslinked spider silk proteins. Moreover, **a mild dry process** was developed to effectively remove the residual

water molecules and ensure the integrity and stability of the fabricated nanostructures.

(4) In Fig. 2a-c, it is true that a higher energy beam can penetrate deeper. However, I cannot understand why e-beam hits and energy dissipation have maxima, not exponential decays, with dependences on the e-beam energy. Did the authors shift the focal points as changing the acceleration voltages? It requires a clear explanation on it.

Response: Thank you for your suggestion. **The focal points were shifted as changing the acceleration voltages.** The energy distribution of electron trajectories describes the probability of observing a total electron energy within a depth slice at z during the scattering process. However, energy distribution is only an indirect characterization of the electron-resist interaction: The exposure process is not due to the absolute energy level (at least not directly) but largely the energy release of the scattering electrons, which in Monte-Carlo simulations (e.g., CASINO) can be "numerically measured" as the energy dissipation of active electrons before and after scattering events.

Although CASINO has built-in functionality for counting electron energy histogram over depth, we would like to emphasize that, dissipation distribution is not equivalent to the gradient magnitude of energy distribution, especially near the maximum penetration depth where electron trajectories no longer align with the z -direction. Hence in Fig. 2a-b what we have shown is a post-processing of the raw data output from CASINO. In short, for each electron trajectory, we collect the incremental energy dissipation between two adjacent nodes where a scattering event happens. This dissipation is then distributed into all depth slices which the trajectory segment penetrates through. All simulation parameters were kept fixed except the beam energy.

Dissipation curves in Fig. 2b attain their maxima right after the peaks of the corresponding energy curves. This observed behavior is in fact plausible. It is consistent with the intuition that electron energy per depth should drop drastically as the majority of electrons start to lose their energy (as indicated by the energy curves). These curves and their peaks in Fig. 2b serve as strong numerical evidence that proves the point of our work: the process of energy release during a electron beam exposure is not a gradual but rather a concentrated phenomenon over depth. In other words, these peaks in the dissipation

curves imply that the largest release of electron energy due to collisions is likely to occur some distance beneath the incident surface.

(5) Detailed descriptions for Fig. 4 are required.

Response: Thank you for your suggestion. We have added the detailed descriptions for Fig. 4. The light green, bright green and dark red fluorescence in nanofishes represented the low power, full power and loss power states in kinetics. Under the NIR-light illustration, the nanofishes embedded with gold nanoshells (peak absorbance at 980 nm) exhibited stronger infrared absorption than the background. With the increase of the temperature from 20 °C to 50 °C, papain (pre-loaded in the nanofishes) was activated (optimum temperature 50 °C - 60 °C) and led to the degradation of spider silk protein. Pepsin is inactive and active in the neutral and acidic environments, respectively. With the decrease of pH from 7.0 to 5.0 and 3.0, pepsin was gradually activated and broke down spider silk proteins into smaller peptides. It was suitable for the controllable drug delivery and release in the digestive systems of humans and many other animals. (Page 9, line 245)

(6) In Fig. 4c, NIR-light off image is not comparable to NIR-light on image. Are two the same image?

Response: Thank you for your comments. **NIR-light off image and NIR-light on image are two images in different imaging-capturing modes of one same sample.** The NIR-light off image was collected from the visible-light channel of the IR microscopy in a gray-scale mode for localizing the position and providing the microstructural information of the sample. It is a general set-up both in the Vistec and Olympus microscopy systems which are widely used. NIR-light on image exhibited the infrared absorption of the sample in the corresponding region under the illumination of NIR-light. In order to clearly distinguish the difference between the nanofishes and background, we selected the heat-map mode for image presentation.

Responses to the comments of Reviewer #2:

This manuscript reports fabrication of 3D structures via voltage modulated electron beam lithography of silk-based films. The dependence of the penetration depth on the acceleration voltage forms the basis of such patterning approach. Functional nanoscale structures can be obtained by modulating the silk-based resists. The work is certainly interesting and can be considered for publication in Nature Communications. The authors should address the following issues.

Response: Thanks for the positive comments.

1) An important metric for lithography is the minimum pitch that can be fabricated. Please report the minimum pitch that can be fabricated with this approach. It will be highly useful to see an array of pillars at such pitch values.

Response: Thank you for your suggestion. We have added the SEM image of an array of nanopillars fabricated with this approach in Supplementary Figure 9a. In order to investigate the performance of spider silk protein in high aspect ratio 3d nanostructuring, the thickness of the protein layer was set above 500 nm, a quite significant value for nanolithography (generally, thicker the photoresist, harder for fine nanopatterning). Finally, the diameter of the obtained nanopillars was 108 nm. **The corresponding minimum pitch reached 198 nm.**

Supplementary Figure 9. The pitch (a), line edge roughness (b) and aspect ratio (c) of the nanostructures fabricated using 3d EBL. LER: line edge roughness.

2) It appears that there is certain degree of line edge roughness in the fabricated structures. Please report line edge roughness values for both in plane and out of plane structures.

Response: Thank you for your suggestion. We have added the line edge roughness values for nanostructures in Supplementary Figure 9b. **The out-of-plane line edge roughness** of the nanopillar (lying on the substrate, mean diameter: 108 nm) was **1.1 nm**. **The in-plane line edge roughness** of the nanowall (mean linewidth: 73 nm) was **1.3 nm**.

3) For a pillar like structure, what is the highest achievable aspect ratio?

Response: Thank you for your comments. We have added the SEM image of the spider silk protein nanopillar in Supplementary Figure 9c. The diameter and height of the nanopillar was 108 nm and 552 nm, respectively. **The highest achievable aspect ratio was 5.11 for a pillar like structure**. In addition, if the pillar structure was changed to the line structure (with enhanced stability arising from the joint supporting), the achievable aspect ratio rose to 19.3 (Supplementary Figure 8).

4) Some sections (brain tumor cells, images in part c and e) of Figure 3 are not adequately described/discussed in the text.

Response: Thank you for your suggestion. We have added the detailed description and discussion in the text. Fig. 3c shows the X-Y plane controllable configuration of spider nanowebs for PMMA nanospheres (~250 nm in diameter) adhesion. The images in the upper row and lower row were the schematics and the corresponding samples, respectively. We fabricated the radial line (red fluorescence) and spiral line (green fluorescence) in sequence to define the materials and functions of the nanowebs in the X-Y plane. When the nanowebs were immersed into the suspension of PMMA nanospheres (~250 nm in diameter), more nanospheres (blue, false color) were adhered to the spiral line (green, false color) than the radial line (red, false color) with a ratio of 34:16. Fig. 3e shows the Z direction controllable configuration of spider nanowebs for single brain tumor cell (U251 glioma cell, ~30 μm in diameter) adhesion. As shown in the schematics (upper row), we fabricated the substrate (red fluorescence) and top layer

(green fluorescence) of the nanowebs in sequence (lower row) to form a single cell scaffold with a diameter matching the tumor cell size. Then, the spider silk nanowebs and U251 glioma cells were placed into 6-well plate. After cell-culture as described in the Methods section, single brain tumor cell (blue fluorescence) successfully adhered and grew on the nanowebs. (Page 7, line 191)

5) Similarly, some sections (NIR activation, heat and pH driven transitions) of Figure 4 are not adequately described/discussed in the text.

Response: Thank you for your suggestion. We have added the further description and discussion in the text. Under the NIR-light illustration, the nanofishes embedded with gold nanoshells (peak absorbance at 980 nm) exhibited stronger infrared absorption than the background. With the increase of the temperature from 20 °C to 50 °C, papain (pre-loaded in the nanofishes) was activated (optimum temperature 50 °C - 60 °C) and led to the degradation of spider silk protein. Pepsin is inactive and active in the neutral and acidic environments, respectively. With the decrease of pH from 7.0 to 5.0 and 3.0, pepsin was gradually activated and broke down spider silk proteins into smaller peptides. It was suitable for the controllable drug delivery and release in the digestive systems of humans and many other animals. (Page 9, line 249)

6) What is the achievable level of velocity for the nanofish?

Response: Thank you for your comments. We have added the velocity test of the spider silk protein nanofish in Supplementary Figure 14. The results showed that the nanofish swam ~240 μm within 3 s. **It means the achievable velocity for the nanofish (20 μm in length) is ~80 μm (four times the length of nanofish) per second.** Furthermore, the self-propelled nanofish was essentially actuated by the energy conversion that transformed biofuel (glucose) to kinetic energy (O₂ bubbles) in a glucose-containing environment at the human physiological level in this work. Glucose oxidase firstly catalyzed the oxidation of glucose to hydrogen peroxide, then catalase further catalyzed the decomposition of hydrogen peroxide to oxygen. Therefore, it is a promising way to

improve the concentrations of enzymes (embedded in the nanofish) and chemical fuel (existing in the environment) for gaining a higher velocity of nanofish.

Supplementary Figure 14. The velocity of the spider silk protein nanofish.

REVIEWERS' COMMENTS

Reviewer #1 (Remarks to the Author):

Authors clearly respond to all reviewer's concerns. I recommend its publication in Nature Communication.

Reviewer #2 (Remarks to the Author):

The authors have addressed my comments. The manuscript can be accepted for publication in Nature Communications.

Responses to the referees' comments: Manuscript ID NCOMMS-21-09897

Responses to the comments of Reviewer #1:

Authors clearly respond to all reviewer's concerns. I recommend its publication in Nature Communication.

Response: Thanks for the very positive comments.

Responses to the comments of Reviewer #2:

The authors have addressed my comments. The manuscript can be accepted for publication in Nature Communications.

Response: Thanks for the very positive comments.